# Placental Syndromes—A New Paradigm in Perinatology

**DOI:** 10.3390/ijerph19127392

**Published:** 2022-06-16

**Authors:** Katarzyna Kosińska-Kaczyńska

**Affiliations:** 2nd Department of Obstetrics and Gynecology, Center of Postgraduate Medical Education, 01-809 Warsaw, Poland; katarzyna.kosinska-kaczynska@cmkp.edu.pl

**Keywords:** placental syndrome, preeclampsia, fetal growth restriction, placental abruption, hypertension, pregnancy, HELLP syndrome, miscarriage

## Abstract

Placental syndromes include pregnancy loss, fetal growth restriction, preeclampsia, preterm delivery, premature rupture of membranes, placental abruption and intrauterine fetal demise. This paper discusses the common etiopathogenesis of those syndromes and the role of angiogenic biomarkers in their development. Pregnancy implantation, placental development and maternal adaptation are complex processes in which fetal and maternal cells interact. The syncytiotrophoblast, trophoblast, uterine natural killer cells and regulatory T cells interfere and interact in all the above-mentioned processes. The proper angioneogenesis and vasculogenesis of the placenta, as well as maternal circulatory adaptation, are dependent on angiogenic factor expression. Insufficient maternal immunotolerance, dysregulation in uterine natural killer or regulatory T cell function, syncytiotrophoblast and trophoblast ischemia and hypoxia or impaired balance in angiogenic factors are all related to the occurrence of placental syndromes. Differences in the time of impairment onset and its intensity and correlation with other dysfunctions result in the development of a specific syndrome. The clinical manifestations in the form of a combination of specific symptoms determine the diagnosis. However, they are just symptoms of an underlying complex trophoblast disorder.

## 1. Introduction

Human placentation in a unique and complex process. It is associated with intensive adaptation in the maternal body, which allows the pregnancy to be maintained and to develop. Adaptation includes changes in the maternal circulation and pulmonary, secretory and immune system changes. These complex mechanisms, dependent on each other, are essential to allow the creation of the placenta and its functioning. Preeclampsia (PE) was the first pregnancy disorder whose etiopathogenesis was connected to abnormal placentation. In 1992, Christopher W. G. Redman proposed the two-stage hypothesis of PE development: earlier placental stage (deficient placentation) and later maternal symptoms (evolving from systemic vascular inflammation) [1]. Roberts and Redman assumed that factors inducing maternal endothelial dysfunction were the link between the two stages [2]. Those factors had an angiogenic function and included vascular endothelial growth factor (VEGF) and placental growth factor (PlGF)—both with proangiogenic functions—and soluble decoy receptor for VEGF and PlGF (named soluble fms-like tyrosine kinase-1 or sFlt) and soluble endoglin (sEng)—with antiangiogenic functions [3]. Nowadays, we know that impaired placentation and disturbed maternal adaptation may lead to several pregnancy complications. In 2019, in a paper by Anne Cathrine Staff, they were called placental syndromes, as they all arise from impaired trophoblast invasion or trophoblast stress [4]. They include pregnancy loss, fetal growth restriction (FGR), preeclampsia (PE), preterm delivery, premature rupture of membranes (PROM), placental abruption and intrauterine fetal demise [4]. This paper will discuss the common etiopathogenesis of all those syndromes and the role of angiogenic biomarkers in their development.

### 1.1. Implantation and Placental Development

The implantation process begins when the blastocyst hatches from the zona pellucida. Whereas the preimplantation conceptus can survive and develop without active maternal nutrition, implantation is the first stage of cooperation between maternal cells and the blastocyst [5]. The trophoblast cells attach to the receptive uterine epithelium, allowing the adhesion of the blastocyst to the decidua. Then, the whole conceptus is drawn into the decidua [6]. Cytotrophoblast cells begin to merge and form a multinuclear layer called the syncytiotrophoblast. Initially, it is formed on the side of blastocyst adhesion. However, when the entire conceptus is drawn into the endometrium, the syncytiotrophoblast surrounds the blastocyst entirely. It produces lytic enzymes and secretes proapoptotic factors, which play a crucial role in implantation [7]. The syncytiotrophoblast invades the decidual stroma and glands. Its proteolytic enzymes destroy the decidua and maternal vessels located inside. At this early stage, the lacunae are formed within the syncytiotrophoblast [6]. They tend to enlarge and connect with each other, and after fusion with maternal vessels, they allow maternal blood to enter the lacunae system (Figure 1). At the late lacunar stage, the cytotrophoblast is mostly located next to the fetal mesenchyme and begins to form the primary villi. The villi expand and reach toward the fusing lacunae filling with maternal blood. Simultaneously, the extravillous trophoblast breaks through the syncytiotrophoblast and forms the cytotrophoblast columns, which migrate into the decidua [6]. The trophoblast invades maternal spiral arteries [7]. At first, it infiltrates the arteries and plugs them to protect the sensitive conceptus from the harmful effect of high oxygen concentration. When the recanalization of the spiral arteries takes place, it starts from the shallowest placentation site (opposite the umbilical cord attachment). Sensitive villi located there are subjected to an increasing concentration of oxygen and oxidative stress, which leads to their atrophy. Afterward, the spiral arteries are unblocked at the site of the deepest placentation (the site of umbilical cord attachment). However, villi developing there are more mature at the time and more resistant to high oxygen concentration and, therefore, they are able to develop further. They will form the placental plate afterward [3].

The fetal mesoderm migrates into the villi forming the secondary villi. They increase in size and ramify. They are drawn into the syncytiotrophoblast layer with an increasing system of fusing lacunae inside (Figure 2). Meanwhile, the extravillous trophoblast penetrates the whole depth of the decidua and into the myometrium until clumps of cytotrophoblast cells fuse and form multinucleated giant cells [6]. It invades deep into the spiral arteries, reaching the depth of the inner myometrium. The process of spiral artery remodeling takes place.

The spiral arteries are invaded by the extravillous trophoblast in an interstitial and endovascular way. The trophoblast penetrates deep into the myometrial segments of the arteries. After initial plugging and subsequent revascularization, the invasive trophoblast removes the endothelial cells and the vascular smooth muscle cells. Trophoblast cells replace the endothelium [8]. Different and complex mechanisms are involved in the remodeling process and include vascular cell de-differentiation, migration, changes in cellular adhesion and sensitivity to death-inducing stimuli [8]. When the process is finished, the spiral arteries are changed from low-flow and high-resistance vessels into high-flow and low-resistance ones. They become wide and flabby as their muscular layer is destroyed. Therefore, they are unable to constrict. The blood flow in the arteries is, therefore, constant with low pressure, which allows maternal blood to wash the villi gently [4]. However, endothelial cell replacement with the extravillous trophoblast is temporary. Re-endothelialization with maternal endothelial cells probably takes place later in pregnancy [9].

As the branches of the umbilical vessels grow into the mesodermal villi, the chorionic villi become vascularized. They are called the tertiary villi. They are covered with a single layer of trophoblast cells and syncytiotrophoblast lying above them. The surface of the syncytiotrophoblast is covered by the microvilli (>1 billion/cm^2^ at term), which increases the total surface area of the placenta. Villi are dipped in the maternal blood filling the intervillous space, supplied by the spiral arteries. The intervillous space is formed by the fusion of all the lacunae of the syncytiotrophoblast (Figure 3). As villi have direct contact with the maternal blood, the human placenta is called hemochorial.

The adequate adaptation of the maternal immune system is essential for pregnancy development. As the conceptus presents both maternal and paternal antigens, maternal immune tolerance is necessary to allow a robust placenta. Maternal decidual regulatory T cells (Tregs) are responsible for inducing immune tolerance and have an anti-inflammatory function [10,11]. Seminal fluid containing cytokines and chemokines induces Tregs to interact with paternal antigens and creates memory Tregs [10]. If the memory is efficient, Tregs induce maternal immunotolerance during implantation and, despite the expression of paternal antigens, the blastocyst is not rejected.

The maternal immune system plays an important role in spiral artery remodeling. Uterine natural killer (uNK) cells take part in the implantation and the remodeling of the arteries. uNK cells are the dominant immune cell type in the decidua. They constitute almost 70% of all mononuclear cells and infiltrate the spiral arteries and endometrial glands [12]. CD56^bright^CD16^−^ constitute the majority of uNK cells, while CD56^dim^CD16^+^ are found in the peripheral blood [13]. Although uNK cells accumulate in large numbers at the implantation site, their role in implantation is still unknown. They may provide memory to aid in vascular remodeling of the placenta during subsequent pregnancies. Interestingly, uNK cells lose their killing phenotype during pregnancy. It was hypothesized that they control decidual vascular function through the secretion of angiogenic growth factors. uNK cells produce VEGF C and PlGF [14]. VEGF C supports endovascular processes and is responsible for the non-cytotoxic activity of uNK cells [15]. Conversely, PlGF plays a role in uNK cell proliferation [16]. uNK cells contribute to spiral artery remodeling and extravillous trophoblast invasion into the decidua and vessels [12]. They interact with the trophoblast cells mainly through the killer-cell immunoglobulin-like receptor (KIR). KIRs, which are expressed on uNK cells, interact with MHC class I molecule proteins expressed on the trophoblast cells invading the decidua and arteries [4]. HLA-C is the only polymorphic MHC molecule expressed by the trophoblast. It is the major ligand for KIRs [17]. uNK KIRs can bind to fetal HLA-C molecules and recognize fetal trophoblast cells. KIRs show polymorphism and are usually grouped into A and B haplotypes. KIR A haplotypes are mostly inhibitory and KIR B haplotypes are mostly activating [18]. The individual haplotype of KIRs and its interaction with HLA-C on the trophoblast cells is essential in proper implantation and uterine artery remodeling.

Three main decidual NK cell subsets (dNK1, dNK2 and dNK3) were identified by RNA-sequencing. The dNK1 cells express higher levels of inhibitory KIRs (KIR2DL1, KIR2DL2 and KIR2DL3) and activatory KIRs (KIR2DS1 and KIR2DS4). dNK1 and dNK2, but not dNK3, express killer cell lectin-like receptor C1, C2 and C3. The dNK1 cell subset expressed higher levels of KIRs and leukocyte immunoglobulin-like receptor B1 that bind HLA-C and HLA-G molecules, respectively, expressed on extravillous trophoblast [19].

The syncytiotrophoblast is the main source of PlGF. PlGF is a member of the VEGF family. It is essential for trophoblast growth and differentiation, as well as vasculogenesis and angioneogenesis in the placenta. It is also secreted into the maternal circulation, where it increases endothelial cell proliferation. The binding of PlGF and VEGF to the endothelial receptors VEGFR-1 and VEGFR-2 promotes vasodilation and the maintenance of the maternal endothelium [20]. It plays a key role in maternal circulatory system adaptation to pregnancy.

Pregnancy implantation, placental development and maternal adaptation are complex processes in which fetal and maternal cells interact. The syncytiotrophoblast, trophoblast, uNK cells and Tregs interfere and interact in all the above-mentioned processes [12]. The proper angioneogenesis and vasculogenesis of the placenta, as well as maternal circulatory adaptation are dependent on angiogenic factor expression. During the process, several errors may occur, disturbing it and leading to the development of placental syndromes.

### 1.2. Early Pregnancy Loss

Immune tolerance at the maternal–fetal interface is essential for pregnancy development. The conceptus is a kind of a semi-allogeneic transplant and, therefore, the adaptation of the maternal immune system is required for pregnancy to be maintained without rejection. The switch from a Th1 to a Th2 cytokine profile plays a key role. Progesterone increases the production of Th2 type cytokines and suppresses the production of Th1 and Th17 cytokines in vitro. TNF alfa, IFN gamma (Th1 type cytokines) and IL-17 (Th17 type cytokine) have embryotoxic and anti-trophoblast actions. IL-17 and IL-22 produced by Th17 cells may be responsible for the rejection of the HLA-C-expressing trophoblast. Immune defects concerning both uNK and Treg cells may contribute to disturbed implantation and early pregnancy loss.

CD4^+^CD25^+^ Tregs are essential for maternal immunotolerance [11]. Both the peripheral blood and uterine Tregs increase in response to implantation [11]. The recognition of paternal/fetal antigens by Tregs is essential to induce immune tolerance. Tregs expressing forkhead box protein 3 (FoxP3) produce immunosuppressive cytokines IL-10 and TGF-β [21]. Female dendritic cells recognize fetal antigens and induce the transformation of effector CD4 T cells into induced Tregs [22]. The induced Tregs are recruited to the decidua. It was suggested that early semen/seminal fluid-induced Tregs were the first present in the decidua during the implantation [12]. Their poor development and recruitment may lead to implantation failure. After implantation, the conceptus induces a subsequent population of Tregs, and they appear later during pregnancy [12]. A decreased Treg population was shown in the decidua in cases of miscarriage with a normal fetal chromosome set. A decreased rate of Tregs was observed in the decidua of women with recurrent pregnancy losses [23].

uNK cells also play a role in immune tolerance development. Elevated uNK cells in material extracted by endometrial scratching were found in women with recurrent pregnancy losses [24]. uNK inhibitory receptors (KIR AA haplotype) were reported to lead to an increased risk of early pregnancy loss when the fetus presented HLA-C2 allotype [25]. Such a combination was associated with inadequate uNK function and impaired trophoblast invasion. The CD56^bright^CD27^+^ subset of uNK cells produces IFN-γ, which dampens Th17 cells and promotes immune tolerance [26]. It was suggested that IL-17, produced by Th17 cells, induced extensive local inflammation and might impair early pregnancy. Women experiencing recurrent pregnancy losses may have decidual IL-10 deficiency [27]. IL-10, produced by uNK, inhibits Th17 cell function. Therefore, it is possible that early pregnancy loss may be caused by a local inflammation derived from dysregulated uNK cells [12]. As decreased Tregs may also promote Th17 function, their insufficient number may play a role in early pregnancy loss [12]. Another interesting cytokine is IL-18. It can induce either Th1 or Th2 response depending on the local cytokine level. IL-18 upregulation was found in the decidua of women with recurrent miscarriage [28]. However, future studies are needed to identify all relations between immune cells and early pregnancy losses prior to proposing the use of uNK, Treg or specific cytokines as biomarkers [29].

Another crosstalk between trophoblast and uNK cells is autophagy. It is a catalytic process that maintains homeostasis in eukaryotic cells, including the degradation of damaged macromolecules and cytoplasmic components of organelles. It has been reported that autophagy can induce immune tolerance at the maternal–fetal interface. The level of autophagy in trophoblasts was found to be decreased in patients with recurrent pregnancy losses. In trophoblasts, it restricts the cytotoxicity of uNK cells; therefore, uNK cells regulate the proliferation and invasion of trophoblasts and damage the trophoblast invasion possibly by targeting insulin-like growth factor 2 [30]. A recent study reported a relationship between autophagy and ATP-binding cassette transporters, which mediate cholesterol transfer across the placenta. Decreased expression of ATP-binding cassette transporters contributes to reduced cholesterol export to the fetus and was observed in women with early miscarriages [31]. Another possible mechanism of relation between inflammatory processes and miscarriage concerns omentin. Omentin is an anti-inflammatory peptide mainly expressed in stromal vascular cells of visceral adipose tissue. It was also found in the placenta. Omentin inhibits TNF alfa and Il-6. Elevated concentrations of omentin were found in the plasma of women with recurrent pregnancy losses [32].

### 1.3. Early-Onset Preeclampsia

The pre-conceptional stage of PE concerns the memory Treg cell generation. It evolves during the exposure to paternal antigens contained in the semen or due to previous conception with the same father. PE was observed to be more common in women with a short period of sexual cohabitation before the pregnancy. The short period may be insufficient to generate Treg memory essential to promote immunotolerance during implantation [33]. Inadequate Treg memory may also be related to long inter-pregnancy periods, as it declines over time [34]. Treg reduction or dysfunction, leading to Treg/Th17 imbalance, was observed in preeclamptic gestation [35,36]. The same mechanism appears to increase the risk of PE in pregnancies conceived via assisted reproductive technologies with donor sperm [37]. uNK cells use KIRs to recognize the fetal HLA-C molecules on invading trophoblast. It was demonstrated that the combination of maternal KIR AA genotypes and fetal HLA-C alleles encoding the C2 epitope was associated with a higher risk of PE [38].

In early pregnancy, the extravillous trophoblast plugs spiral arteries to maintain hypoxia in the placentation process, which promotes trophoblast proliferation rather than differentiation [39]. If the plugging is insufficient or resolves too early, the trophoblast cells are submitted to high oxygen concentration and oxidative stress. The primary villi presented to high oxygen concentrations become atrophic. These processes lead to pregnancy loss or PE and placental insufficiency.

The proper remodeling of spiral arteries is essential for placental development and function. If trophoblast invasion is not deep enough, the process is dysfunctional. If any of the steps in remodeling are insufficient, including uNK and extravillous trophoblast cell function, the arteries do not turn into high-flow low-resistance vessels but remain low-flow and high-resistance. They maintain the functioning of the smooth muscle layer and react to stimuli with contraction. Therefore, maternal blood flowing through these vessels into the intervillous space with high velocity and high pressure is likely to injure the chorionic villi, both hydrodynamically and biochemically. The syncytiotrophoblast and trophoblast are subjected to hypoxia during vasoconstriction and to oxidative stress after reperfusion. The pulsatile flow with higher pressure generates ischemia–reperfusion injury and placental stress [4]. Maternal obesity and chronic vascular inflammation were also shown to be independent risk factors of poor spiral artery remodeling [40]. The syncytiotrophoblast is the first layer exposed to the injury, but the villi are entirely damaged as well. Placental malperfusion leads to placental endoplasmic reticulum (ER) stress and the activation of the unfolded protein response [41]. Unsuccessful folding blocks protein synthesis and is related to the development of a small placenta in early PE and FGR [4]. The syncytiotrophoblast releases lower amounts of PlGF, which also impairs placental angioneogenesis and vasculogenesis. The placenta is unable to develop properly and becomes insufficient.

Decidual inflammation may lead to decidual acute atherosis. Arterial foam cell lesions resemble the early stages of atherosclerosis. Such changes were observed in preeclampsia [42,43]. The obstruction of spiral arteries with atherosclerotic plaques and thrombosis results in placental malperfusion [43].

Trophoblast invasion is also regulated by an extracellular-matrix-associated protein mainly expressed by the syncytiotrophoblast called placental protein 5 (PP5)/tissue factor pathway inhibitor-2 (TFPI-2). It was found that increased placental PP5/TFPI-2 expression may be associated with abnormal placentation in early preeclampsia, with or without HELLP syndrome [44].

The syncytiotrophoblast layer becomes disintegrated under hypoxic stress. It releases debris with multinuclear fragments and microvesicles (sized 100–1000 nm) [45,46]. Syncytiotrophoblast microvesicles (STBMs) carry relevant molecular cargos. They express heat shock protein 70 and high-morbidity group box 1, which have a proinflammatory action. The tissue factor located in STBM membrane has a procoagulant activity, while Flt-1 and endoglin contribute to maternal endothelial dysfunction [45]. STBMs were also shown to suppress NK and T cell responses in vitro, possibly due to their expression of MICA/B and Fas ligand, respectively [47]. All those STBM activities trigger maternal endothelial dysfunction and are a key factor in PE development. In vitro studies demonstrated that STBMs prepared from the placenta caused endothelial cell dysfunction [48]. The plasma of preeclamptic women was found to be characterized by higher concentrations of STBMs [46].

The syncytiotrophoblast exposed to stress produces lower amounts of PlGF and higher amounts of sFlt-1. The expression of Flt-1 is largely located in the abnormal clusters of degenerative syncytiotrophoblast, known as syncytial knots [49]. sFlt-1 is a soluble receptor for VEGF and PlGF [20]. The syncytiotrophoblast is the dominant source of sFlt-1. VEGF binds to sFlt-1 and becomes inactive. In preeclamptic women, low serum concentrations of PlGF and/or high concentrations of sFlt-1 are observed [50]. A high circulating level of sFlt-1 contributes to maternal endothelial dysfunction [49]. It leads to a systemic inflammatory response, which involves the leukocyte and complements activation, insulin resistance and hyperlipidemia [51]. sFlt-1 is shed strictly into the maternal blood and accesses the maternal circulation through the uterine veins. PE symptoms are related to endothelial dysfunction caused by STBM, sFlt-1 and sEng. If the placental debris causes endothelial injury in the pulmonary circulation, pulmonary edema or adult respiratory distress syndrome may develop. As regards the central nervous system, visual disturbances, headache, eclampsia, intracranial hemorrhage or cerebral infarction may occur. Blurred vision or retinal detachment may be observed. Kidney dysfunction, proteinuria or liver dysfunction, liver hematoma and hepatic rupture result from the endothelial dysfunction of the organs. The impaired function of the hemopoietic system may cause hemolysis, leukocytosis and thrombocytopenia. As regards the cardiovascular system, cardiomegaly, aortic dissection or coronary artery spasm may be observed. Less common symptoms of endothelial injury include pancreatitis, intestinal ischemia or necrosis and splenic rupture. General inflammation in the endothelium causes hypertension and intravascular thrombosis. Therefore, PE is thought to be a systemic inflammatory disorder [52]. The sFlt-1/PlGF ratio or PlGF alone with or without clinical characteristics can facilitate second- and third-trimester prediction of early-onset preeclampsia. However, placental angiogenic biomarkers are specific to the current pregnancy and not a reflection of maternal predisposition to PE [53].

A relationship between maternal infections and PE was established. Most studies investigated periodontal disease and urinary tract infections [54]. Recently the maternal microbiome and dysbiosis were hypothesized to play a role in PE etiology. Changes in the composition of human gut microbiota have been reported in women with PE. The gut microbiota stimulates epithelial cells and T cells, which maintain intestinal barrier integrity and commensalism. In the case of dysbiosis, the tightness of the intestinal barrier is lost, and bacteria and the products of their metabolism pass through it excessively. Short-chain fatty acids, produced by the microbiota, induce the formation of pro-inflammatory Th17 lymphocytes and Treg dysfunction. Lipopolysaccharide, which is a component of the cell membrane of Gram-negative bacteria, inhibits Th2 and Th17 responses and influences the differentiation of Treg lymphocytes [55]. Treg reduction or dysfunction, leading to Th17/Treg imbalance, has been observed in cases of abnormal placental implantation. A recent meta-analysis demonstrated that SARS-CoV-2 infection during pregnancy is associated with a significant increase in the risk of developing PE [54]. The pathway may be endothelial dysfunction as SARS-CoV-2 infects endothelial cells that express angiotensin-converting enzyme 2 [52]. Angiotensin-converting enzyme 2 was also found on trophoblast cells. Pregnant women with severe COVID-19 were found to have elevated maternal plasma concentration of sFlt-1 and a high sFlt-1/PlGF ratio, similar to those developing PE [56]. 

### 1.4. Late-Onset Preeclampsia

In late-onset PE, adequate for gestational age fetuses and normal umbilical artery Doppler suggest normal placental function. On the contrary, late-onset or term PE is more common in multiple gestations, pregnancies with large for gestational age fetuses, hyperplacentation or diabetes. In normal pregnancy, the intervillous space gets relatively smaller after 30 weeks of gestation due to the expansion of the chorionic villi. The large placenta and fetus have an increasing demand for oxygen supply. However, when the placenta outgrows the uterine capacity, uterine contractions start to appear more frequently and decrease the intervillous space blood supply during the muscle contraction, the intervillous perfusion becomes insufficient, and the same hypoxic stress occurs in the syncytiotrophoblast as in early-onset PE [4,43]. The same placenta-shed debris is released into the maternal circulation, causing general inflammation and endothelial destabilization. As term PE is unrelated to poor placentation, its prediction in the first trimester of pregnancy is much weaker. However, as hypoxia occurs in the syncytiotrophoblast, the same anti-angiogenic factors are released into the maternal circulation as in early-onset PE [57]. In late-onset PE, the maternal concentration of sFlt-1 is elevated, while PlGF is decreased in comparison with normal gestation [58]. The prediction model of screening using maternal factors, mean arterial pressure, PlGF and sFlt-1 predicts about 85% of term PE, at a screen-positive rate of 10% [59,60,61,62]. Although late-onset PE is associated with a different etiology, it is related to the same syncytiotrophoblast hypoxia and stress, only appearing later in gestation. As the trail of placenta debris shedding appears, the same immune and anti-angiogenic mechanisms come into action, leading to the development of PE symptoms.

### 1.5. HELLP Syndrome

HELLP is an acronym for hemolysis, elevated liver enzymes and low platelets. It occurs in 0.2–0.8% of all pregnancies [63]. It is usually accompanied by hypertension or preeclampsia but about 15% of patients developing HELLP syndrome are normotensive. The majority of studies evaluated angiogenic biomarkers in PE and/or HELLP jointly as the same etiopathogenesis of both syndromes was discussed [64,65]. The most important issue is whether maternal immune responses and the invading trophoblasts generate normal or abnormal placentation [63]. Due to placental hypoxia, anti-angiogenic factors are shed into the maternal circulation and cause endothelial injury, which manifests as impaired liver function, hemolysis and thrombotic microangiopathy. A high sFlt-1 expression was described in the placenta of women who developed HELLP [65]. High sFl-1 and low PlGF concentrations were observed in the maternal blood in HELLP patients. The angiogenic biomarker profile was similar to the one seen in PE [66,67]. B7-H4, a checkpoint molecule of the B7 family, regulates T-cell activation and cytokine secretion. It was found to be decreased in the decidual basalis tissues of early-onset PE and HELLP, suggesting that B7-H4 is involved in the pathogenesis of both disorders [68]. Therefore, HELLP is considered as a placental syndrome.

### 1.6. FGR and Intrauterine Fetal Demise

The etiopathogenesis of FGR is multifactorial. Fetal growth may be restricted due to maternal causes (such as undernutrition or chronic diseases), fetal abnormalities (genetic or metabolic disorders or birth defects) or placental issues (placental or umbilical cord abnormalities and insufficiency). Placental ischemia leads to placental insufficiency. An insufficient placenta is unable to provide the fetus with adequate amounts of oxygen or nutrients, so the fetus cannot achieve its genetic growth potential and becomes growth-restricted. If FGR is related to placental hypoxia, it is usually accompanied by PE as well. Elevated serum concentrations of sFlt-1 and sFlt-1/PlGF ratio are observed in women with PE and FGR [69].

In clinical practice, a criterion of estimated fetal weight below the 10th percentile is usually considered as a definition of small for gestational age fetus (SGA). However, the group of SGA fetuses includes small but healthy children as well as growth-restricted ones. Those with growth restriction may be considered as an adverse neonatal outcome, so it is essential to distinguish between healthy SGA and FGR. However, it may be a huge challenge in clinical practice. A consensus definition for FGR was determined through the Delphi procedure. Early FGR (diagnosed below 32 weeks of gestation) was diagnosed in the presence of three solitary parameters (fetal abdominal circumference <third centile, estimated fetal weight <third centile and absent end-diastolic flow in the umbilical artery) and four contributory parameters (abdominal circumference or estimated fetal weight <10th centile combined with the pulsatility index >95th centile in either the umbilical artery or uterine artery) [70]. The Doppler criteria reflect the etiopathogenesis of placental ischemia and hypoxia in FGR. An abnormal Doppler of umbilical arteries suggests partially damaged chorionic villi and placental insufficiency.

In case of poor placentation and inadequate spiral artery remodeling, the hypoxic stress of the syncytiotrophoblast occurs analogously to PE. A similar decrease in PlGF expression and increase in anti-angiogenic factor release is observed in FGR and in PE. Impaired trophoblast invasion leads to inadequate spiral artery remodeling and subsequent trophoblast hypoxia. The hypoxic cytotrophoblast rather than the syncytiotrophoblast plays a dominant role in FGR pathogenesis. In stressed trophoblast cells, placental ER activates the unfolded protein response, leading to the development of a small insufficient placenta, which is unable to supply the growing fetus. It provides the fetus with insufficient amount of oxygen and nutrition, and the fetus becomes growth restricted. The insufficient supply leads to abnormal growth, hypoxia and subsequent acidosis of the fetus. Acidosis and anaerobic metabolism in cardiac muscle leads to cardiac arrest and fetal death.

Angiogenic biomarkers were found to reflect the placental hypoxic stress in growth-restricted, but not in healthy SGA fetuses. sFlt-1/PlGF ratios were significantly higher in the serum of women with FGR fetuses in comparison with adequate for gestational age and SGA fetuses [71]. The higher the sFlt-1/PlGF ratios observed were, the lower were the gestational age at delivery and time from diagnosis to delivery. The sFlt-1/PlGF ratio was negatively correlated with the time of neonatal care unit hospitalization. The ratios were proved to be well-correlated with the Doppler ultrasound findings and the occurrence of adverse outcomes [71]. Angiogenic factor dysregulation, being analogous to that in PE, suggests the common etiopathogenesis of FGR and PE, demonstrating that FGR is one of the placental syndromes.

### 1.7. Placental Abruption

Placental abruption is defined as the complete or partial separation of the placenta before delivery. The diagnosis of placental abruption is clinical [70]. Its etiology is not completely understood, but several factors evolving from impaired placentation may play a crucial role in abruption. It is directly caused by the rupture of maternal vessels in the decidua and hemorrhage at the decidual–placental interface [72]. The vasoconstriction and thrombosis of the decidual vessels with associated decidual necrosis and venous hemorrhage may be trigger factors for vessel rupture. Similar disturbances may appear in severe decidual atherosis observed in PE. A sudden increase in arterial pressure may cause vessel rupture in hypertension or PE. Therefore, impaired placentation and uteroplacental underperfusion are considered the key mechanisms causing abruption.

Angiogenic factors have been studied in pregnancies complicated by abruption. However, due to the unpredictable occurrence of abruption, such research is difficult to conduct. Therefore, the available data are scarce. A study by Signore at el. revealed that the serum levels of PlGF were decreased and the ratios of sFlt-1/PlGF were increased in nulliparous women before abruption, but only in those who also developed PE [73]. Vallalain et al. studied extremely high values of the sFlt-1/PlGF ratio. A total of 11.4% of women with the ratio exceeding 655 experienced placental abruption later in pregnancy [74]. Other possible biomarkers of abruption are pregnancy-associated plasma protein A, maternal serum alpha fetoprotein and inhibin A [75]. Although the data are limited, it may be hypothesized that abruption may be one of the placental syndromes caused by impaired placentation.

### 1.8. Preterm Delivery and Premature Rupture of Membranes

PE or FGR are often related to preterm delivery for maternal or fetal indications. Preterm, prelabor rupture of the membranes (PPROM) is responsible for 30–40% of preterm deliveries [76]. Maternal placental vascular lesions are present in 34% of patients with preterm delivery and in 35% of patients with PPROM [77]. Matrix metalloproteinases are enzymes degrading extracellular matrix macromolecules, including collagens. Matrix metalloproteinase-9 expression was found in the amnion and chorion. An increased concentration of matrix metalloproteinase-9 was found in the amniotic fluid in spontaneous rupture of membranes at term [78]. It was hypothesized that VEGF might be a mediator for membrane impairment and subsequent rupture due to its ability to activate matrix metalloproteinases through tissue plasminogen activator [79]. Both VEGFR-1 and VEGFR-2 receptors are present in the human amnion [76]. VEGF and VEGFR-1 in the amnion were also suggested to play a role in the pathophysiology of PPROM [79]. The upregulation of their genes was demonstrated in women with rupture of membranes, both term and preterm [58]. The upregulation occurred regardless of the inflammatory status and was present in the amnion and decidua. VEGF and sVEGFR-1 are present in the human amniotic fluid as well [76]. Women with PPROM are characterized by lower amniotic fluid concentrations of sVEGFR-1 regardless of inflammation occurrence [76]. Decreased amniotic fluid concentrations of sVEGFR-1 lead to increased free VEGF concentrations/activity in the amniotic fluid cavity. VEGF induces the activation of matrix-degrading enzymes, leading to membrane rupture [76,80].

Inadequate maternal immune adaptation may lead to maternal anti-fetal rejection analogous to the mechanism of allograft rejection. Maternal lymphocytes can infiltrate the chorioamniotic membranes and lead to chronic chorioamnionitis. Aseptic inflammation causes the activation of metalloproteinases and leads to the rupture of membranes. An overexpression of C-X-C motif chemokine 10 (CXCL10), a marker of allograft rejection, in amniotic fluid was observed in women with preterm delivery [81].

## 2. Discussion

The present paper discusses a hypothesis of placental dysfunction being the physiological control, acting as the biologic clock of pregnancy maintenance. Angiogenic factor concentrations change in the maternal blood during pregnancy. The PlGF level increases from the early pregnancy up to 29–32 weeks of gestation and decreases afterward [82]. It reflects the lower production of PlGF by the syncytiotrophoblast and trophoblast in the third trimester of pregnancy as the placenta gets “older” and more dysfunctional. The concentration of sFlt-1 remains stable for up to 29–32 weeks of gestation and increases afterward; as the placenta gets larger, the chorionic villi become more branched and complex with a higher need for oxygen supplementation and the intervillous space between them gets relatively smaller. When the syncytiotrophoblast experiences lower oxygen supply and as it becomes hypoxic, more sFlt-1 is shed into the maternal circulation [82]. It was shown that PlGF, sFlt-1 and the sFlt-1/PlGF ratio were associated with spontaneous delivery [83,84]. Therefore, the associations between the angiogenetic biomarkers may play a role as a pregnancy biologic clock. It was hypothesized that many weeks post-term all placentas would be so hypoxic and release such high amounts of sFlt-1 that all pregnant women would develop PE then [50]. In case of impaired placentation, placental stress develops early in pregnancy and leads to PE, FGR or PPROM, which are all causes of preterm deliveries. These findings suggest that placental insufficiency may be an indirect trigger of labor.

Nowadays, PE is considered to be a syndrome with multiple underlying causes and etiologies rather than a disease in itself [52]. According to Jung et al. multiple pathologic processes may contribute to PE. They include uteroplacental ischemia, bacterial infection, SARS-CoV-2 infection, maternal intestinal dysbiosis, sleep disorders, hydatidiform moles, fetal diseases, autoimmune disorders, placental aging, breakdown of maternal–fetal immune tolerance and endocrine disorders [52]. In this review, the common etiopathogenesis of placental disorders is discussed. Insufficient maternal immunotolerance, dysregulation in uNK or Treg function, syncytiotrophoblast and trophoblast hypoxia or impaired balance of angiogenic factors are all related to the occurrence of placental syndromes. Differences in the onset of impairment and its intensity and relation to other dysfunctions result in the development of a specific syndrome such as pregnancy loss, FGR, PE, preterm delivery, PROM, placental abruption or intrauterine fetal demise. Clinical manifestations in the form of a combination of specific symptoms determine the diagnosis. However, they are just symptoms of an underlying complex trophoblast disorder.

## 3. Conclusions

Placental syndromes include pregnancy loss, FGR, PE, preterm delivery, PROM, placental abruption and intrauterine fetal demise. They are all symptoms of insufficient maternal immunotolerance, dysregulation in uNK or Treg function, syncytiotrophoblast and trophoblast hypoxia or impaired balance of angiogenic factors.

## Figures and Tables

**Figure 1 ijerph-19-07392-f001:**
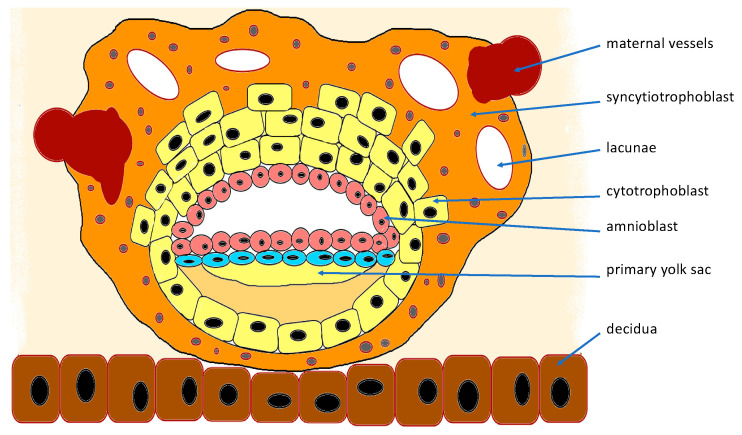
Scheme of 9th day of implantation, implanted blastocyst.

**Figure 2 ijerph-19-07392-f002:**
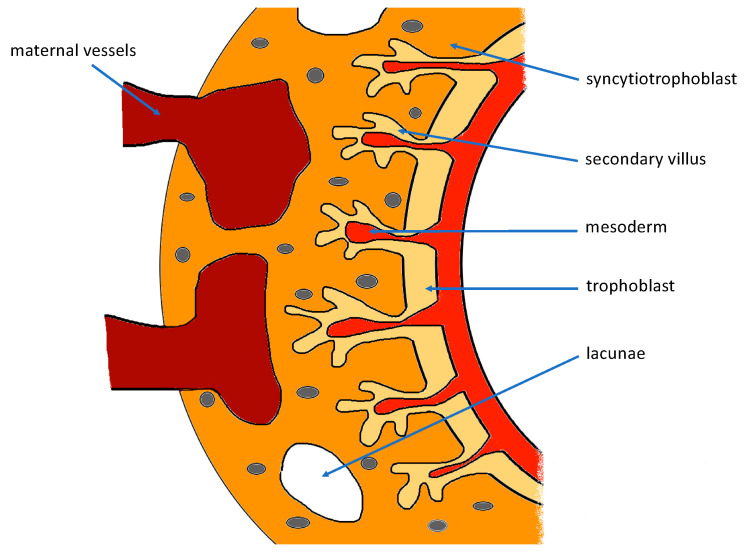
Scheme of placental development—secondary chorionic villus formation.

**Figure 3 ijerph-19-07392-f003:**
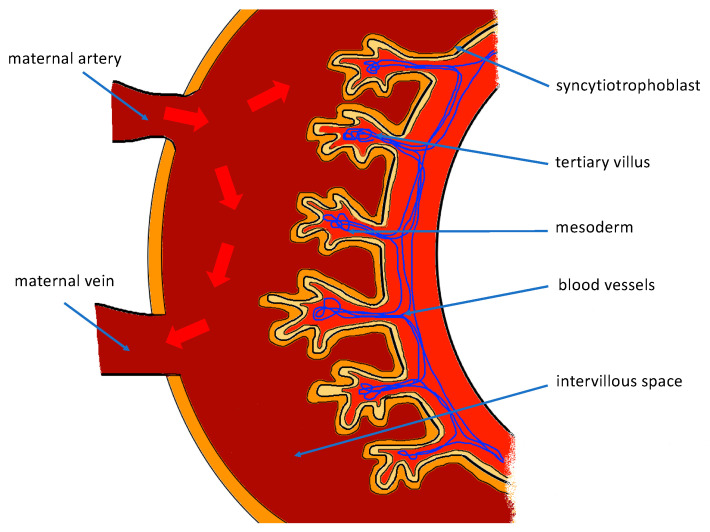
Scheme of placental development—tertiary chorionic villus formation.

## Data Availability

Not applicable.

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
