# Peer review of "Placental Syndromes—A New Paradigm in Perinatology"

_ijerph, 2022, doi:10.3390/ijerph19127392_

Round 1

Reviewer 1 Report

Comments and suggestions for the author

The idea of conducting a review summarizing placental syndromes is current.

-       Line 127 uNK cells may provide memory to aid in vascular remodeling of the placenta during subsequent pregnancies. 

-       Please define the uNK cell subsets termed dNK1, dNK2, and dNK3. The dNK1 cell subset expressed higher levels of KIRs and LILRB1 receptors that bind HLA-C and HLA-G molecules, respectively, expressed on extravillous trophoblast. 

-       In the early pregnancy loss section - repeated implantation failure, or recurrent pregnancy loss DOI: 10.1007/s00404-020-05679-z

-       Future studies should serve as a prerequisite prior to proposing the use of uNK as a biomarker or before targeting uNK cells for therapeutic purposes addressing RIF and RM. DOI: 10.3390/biomedicines9101425

-       Lines 187-188 a possible mechanism - Induction of pregnancy loss by activation of deciduous cytotoxicity and inhibition of trophoblast invasion is secondary to suppressing trophoblastic cell autophagy. DOI: 10.1186/s12964-020-00579-w

-       Furthermore, dysregulated autophagy combined with increased oxidative toxicity and aberrant expression of placental ATP-binding cassette transporters affects materno-fetal health in early pregnancy loss. DOI: 10.3390/antiox10111742

-       Line 306 growth restriction and no mechanisms for intrauterine fetal death discussed

Some suggestions: 

-       the role of omentin, an anti-inflammatory adipokine in early pregnancy loss

-       a significant upregulation of IL-18 in the decidua was restricted to patients with recurrent miscarriage

-       Progesterone increases the production of Th2-type cytokines and suppresses the production of Th1 and Th17 cytokines in vitro. Th1-type cytokines TNF-α, IFN-γ, and Th17 IL-17 have embryotoxic and anti-trophoblast actions. IL-17 and IL-22 produced by Th17 cells may be responsible for the rejection of the HLA-C-expressing parent trophoblast.

-       Systemic transcriptome in early- and late-onset pre-eclampsia shows distinct pathology and novel biomarkers (EBI3, IGF2, ORMDL3, GATA2 and KIR2DL4) expressed specifically in extravillous trophoblast and syncytiotrophoblast.

-       PIK3CB, FLT1, CBLC and ITGA7 are the core regulatory genes differentially expressed in early-onset pre-eclampsia decidua compared with late-onset preeclampsia.

-       sFlt-1/PlGF ratio or PlGF alone with or without clinical characteristics can facilitate second-/third-trimester prediction of early-onset and late-onset preeclampsia

-       B7-H4, regulates T-cell activation, cytokine secretion, tumor progression, and invasion capacities, and is involved in the pathogenesis of PE and HELLP

-       increased placental PP5/TFPI-2 expression may be associated with abnormal placentation in early preeclampsia, with or without HELLP syndrome

-       The distinct dysregulation of placental protein (PP)1 and PP5 expression in either late-onset or early-onset preeclampsia.

-       The expression of Sonic hedgehog (SHH) protein in placental tissues is associated with the oxidative stress mechanism during preeclampsia

-       systemic free thiols, ischemia-modified albumin, leptin, and sRAGE -  potential biomarkers of pregnancies complicated by combined FGR and PE

-       A high sFlt1/P1GF ratio in an index pregnancy is not associated with a higher risk of ischemic placental disease in a subsequent pregnancy

-       Women with all three abnormal pregnancy-associated plasma protein A, maternal serum alpha-fetoprotein, and inhibin-A were at an 8.8-fold  risk of abruption.

-       Please define the potential biomarkers in every section, or it would be interesting to make a table that includes these biomarkers that define placental syndromes.

 Kind regards

Author Response

Dear Editor,

I thank the Reviewer for the constructive comment. I have revised the manuscript according to the suggestion and I hope that the changes will convince the Reviewer and the Editor that the paper is worthy of publication in International Journal of Environmental Research and Public Health.

 Below are the answers to the Reviewer’s suggestions:

-       Line 127 uNK cells may provide memory to aid in vascular remodeling of the placenta during subsequent pregnancies. 

-       Please define the uNK cell subsets termed dNK1, dNK2, and dNK3. The dNK1 cell subset expressed higher levels of KIRs and LILRB1 receptors that bind HLA-C and HLA-G molecules, respectively, expressed on extravillous trophoblast. 

-       In the early pregnancy loss section - repeated implantation failure, or recurrent pregnancy loss DOI: 10.1007/s00404-020-05679-z

-       Future studies should serve as a prerequisite prior to proposing the use of uNK as a biomarker or before targeting uNK cells for therapeutic purposes addressing RIF and RM. DOI: 10.3390/biomedicines9101425

-       Lines 187-188 a possible mechanism - Induction of pregnancy loss by activation of deciduous cytotoxicity and inhibition of trophoblast invasion is secondary to suppressing trophoblastic cell autophagy. DOI: 10.1186/s12964-020-00579-w

-       Furthermore, dysregulated autophagy combined with increased oxidative toxicity and aberrant expression of placental ATP-binding cassette transporters affects materno-fetal health in early pregnancy loss. DOI: 10.3390/antiox10111742

Thank you for suggesting all the above issues. They were all discussed in the text and all references were added to the manuscript. It helped in improving the manuscript.

-       Line 306 growth restriction and no mechanisms for intrauterine fetal death discussed

The mechanism of placental insufficiency and fetal growth restriction and fetal demise was added to the manuscript and discussed.

Some suggestions: 

-       the role of omentin, an anti-inflammatory adipokine in early pregnancy loss

It was added to the manuscript.

-       a significant upregulation of IL-18 in the decidua was restricted to patients with recurrent miscarriage

The role of IL-18 was added to the early pregnancy loss paragraph.

-       Progesterone increases the production of Th2-type cytokines and suppresses the production of Th1 and Th17 cytokines in vitro. Th1-type cytokines TNF-α, IFN-γ, and Th17 IL-17 have embryotoxic and anti-trophoblast actions. IL-17 and IL-22 produced by Th17 cells may be responsible for the rejection of the HLA-C-expressing parent trophoblast.

This section was added to the manuscript.

-       Systemic transcriptome in early- and late-onset pre-eclampsia shows distinct pathology and novel biomarkers (EBI3, IGF2, ORMDL3, GATA2 and KIR2DL4) expressed specifically in extravillous trophoblast and syncytiotrophoblast.

-       PIK3CB, FLT1, CBLC and ITGA7 are the core regulatory genes differentially expressed in early-onset pre-eclampsia decidua compared with late-onset preeclampsia.

The paper by Gue et al. demonstrates distinct pathology of early-onset and late-onset preeclampsia, which was also discussed in the manuscript. EBI3, IGF2, ORMDL3, GATA2 and KIR2DL4 were experimentally verified with patient blood samples, however further research is needed to verify it they are reliable biomarkers. My paper discusses the common etiopathogenesis of placental syndromes and the role of angiogenic biomarkers in their development. To discuss all the syndromes, it is necessary to be succinct and to mention only the biomarkers of proven efficacy instead of all used in experimental biology. The second mentioned paper by Tong et al. discusses the genes expression differences between the early-onset and late-onset preeclampsia due to their different pathology.

-       sFlt-1/PlGF ratio or PlGF alone with or without clinical characteristics can facilitate second-/third-trimester prediction of early-onset and late-onset preeclampsia

This information was added to the text.

-       B7-H4, regulates T-cell activation, cytokine secretion, tumor progression, and invasion capacities, and is involved in the pathogenesis of PE and HELLP

This information was added to the paragraph on HELLP syndrome.

-       increased placental PP5/TFPI-2 expression may be associated with abnormal placentation in early preeclampsia, with or without HELLP syndrome

-       The distinct dysregulation of placental protein (PP)1 and PP5 expression in either late-onset or early-onset preeclampsia.

This information was added to the paragraph on early-onset PE.

-       The expression of Sonic hedgehog (SHH) protein in placental tissues is associated with the oxidative stress mechanism during preeclampsia

Due to limited nature of the review discussing all placental syndromes only assorted biomarkers can be mentioned. I have chosen the most widely researched.

-       systemic free thiols, ischemia-modified albumin, leptin, and sRAGE -  potential biomarkers of pregnancies complicated by combined FGR and PE

Due to limited nature of the review discussing all placental syndromes only assorted biomarkers can be mentioned. I have chosen the most widely researched. I believe it is not possible to mention all potential biomarkers.

-       A high sFlt1/P1GF ratio in an index pregnancy is not associated with a higher risk of ischemic placental disease in a subsequent pregnancy

It was added to the text.

-       Women with all three abnormal pregnancy-associated plasma protein A, maternal serum alpha-fetoprotein, and inhibin-A were at an 8.8-fold risk of abruption.

 The reference was added to the manuscript.

-       Please define the potential biomarkers in every section, or it would be interesting to make a table that includes these biomarkers that define placental syndromes.

Although it is a great idea, I find it too complicated for this manuscript. The paper concerns all placental syndromes and discusses the common issues in their etiology and the role of angiogenic biomarkers in their development. Taking into account the size of the paper I was not able to mention all the possible biomarkers of the discussed disorders. However, as it would be very interesting, I believe it would a great idea for another manuscript.

Reviewer 2 Report

The work presented by Konsinka tries to highlight different phenomena that together lead to placental dysfunction in abnormal conditions during pregnancy. The review has a precise sequence, and the ideas addressed in it are relevant to the field of reproductive biology. Although the author tries to mention the role of immune cells and angiogenic factors produced by the cellular components of the placenta, I consider it relevant to include other sections where he relates to infectious factors that, as the author correctly cites, are related to placental dysfunction. In addition, The images and figure captions lack quality, which harms the content of the review. Precise figures are suggested here where the processes and components discussed in each section are specified and a figure caption detailing this content. That will give a better presentation of this work. 

Author Response

The work presented by Konsinka tries to highlight different phenomena that together lead to placental dysfunction in abnormal conditions during pregnancy. The review has a precise sequence, and the ideas addressed in it are relevant to the field of reproductive biology. Although the author tries to mention the role of immune cells and angiogenic factors produced by the cellular components of the placenta, I consider it relevant to include other sections where he relates to infectious factors that, as the author correctly cites, are related to placental dysfunction. In addition, The images and figure captions lack quality, which harms the content of the review. Precise figures are suggested here where the processes and components discussed in each section are specified and a figure caption detailing this content. That will give a better presentation of this work. 

The paragraph on maternal infections and their relation with placental dysfunction was added to the manuscript.  I have recreated the figures the best I was able to.

Reviewer 3 Report

The article by Ms Katarzyna Kosińska-Kaczyńska is an interesting attempt to summarize the current state of knowledge regarding the function of the placenta.

Despite all the Reviewer's sympathy and admiration for Assoc Prof Mrs Kosińska-Kaczyńska achievements so far, a few shortcomings should be shown.

Forcing the collective concept: of placental syndromes (PS) is not entirely justified. This definition is seen in the literature but for PE and FGR, possibly early pregnancy loss, and not for preterm labor and PPROM. These states cannot be searched for and related to each other just because similar substances are detected in them. The etiopathogenesis of premature delivery is multifactorial, as mentioned by the Author, but she focuses only on proving her own thesis. If this was the case, why did the term PS not described as placental dysfunctions in DM or GDM and related to fetal programming? eg. Longtine MS, Nelson DM. Placental dysfunction and fetal programming: the importance of placental size, shape, histopathology, and molecular composition. Semin Reprod Med. 2011;29(3):187-196. doi:10.1055/s-0031-1275515

It is understandable that in her work the Author wanted to focus on the immunological processes and angiogenesis occurring in some pathologies of pregnancy, but it seems insufficient to name these conditions described by the Author jointly as placental syndromes.

The placenta is a very important element of the proper course of pregnancy and following this lead, please indicate the pathology of pregnancy of obstetric origin that would have nothing to do with the dysfunction of the placenta :)

Summing up, I suggest changing the title by clarifying the discussed mechanisms occurring in the above pathologies of pregnancy (immunological disorders and angiogenesis in the placenta with the most common pathologies ??- an option, not a title).

Possibly broadening the review to include placental disorders in diabetes.

Author Response

Dear Reviewer, thank you for your suggestions. The term “placental syndromes” was cited from the paper by Anne Cathrine Staff I 2019. The author concluded that the wide spectrum of poor placentation disorders existing with multiple risk factors leads to heterogenous pregnancy complications like: pregnancy loss, recurrent pregnancy abortions, fetal growth restriction, early-onset preeclampsia, preterm delivery and premature rupture of membranes, abruption placenta and intrauterine fetal death. The term “placental syndromes” and its definition was taken from that paper without any changes. I have added this explanation in the manuscript. As all the discussed placental syndromes have origin in poor placentation due to several linked abnormalities, in my opinion gestational diabetes mellitus does not meet this criterion, while complications occurring in pregestational diabetes may lead to placental syndromes, however the etiopathogenesis in these cases is the same as described in the manuscript. Thank you for your kind words but due to my sympathy and admiration for Anne Cathrine Staff I believe citing her term “placental syndromes” id correct without any modifications.

Round 2

Reviewer 1 Report

Dear Author,

I want to congratulate you on your work in writing this article.

Kind regards

Reviewer 3 Report

Review work with quite a pleasant reception. You can see a large amount of work done by the Author. The changes had a positive effect on the overall assessment. I wish you many citations from the above article.